# Early Dynamics of Circulating Tumor DNA Following Curative Hypofractionated Radiotherapy Related to Disease Control in Lung Cancer

**DOI:** 10.3390/diagnostics15101198

**Published:** 2025-05-09

**Authors:** Kyungmi Yang, Jae Myoung Noh, Yeon Jeong Kim, Hongryull Pyo

**Affiliations:** 1Department of Radiation Oncology, Samsung Medical Center, Sungkyunkwan University School of Medicine, Seoul 06351, Republic of Korea; 2Samsung Genome Institute, Samsung Medical Center, Seoul 06351, Republic of Korea

**Keywords:** radiotherapy, cell-free DNA, liquid biopsy, lung cancer

## Abstract

**Background/objectives:** We aimed to characterize the dynamic pattern of circulating tumor DNA (ctDNA) during hypofractionated radiation therapy (RT) in patients with lung cancer and assess its clinical relevance. **Metholds:** Prospectively, 24 patients diagnosed with early-stage lung cancer underwent curative RT with 60–64 Gy in 4–20 fractions. Blood samples were collected at baseline (D0) and on post-RT days 1–3 and 7 (D1–3 and D7). The ctDNA was longitudinally analyzed using LiquidSCAN. To find a feasible index associated with outcome, total VAF(%), max VAF(%), total GE (hGE/mL) and max GE (hGE/mL), were evaluated. **Results:** Thirteen patients with available samples were analyzed with a median 22.2-month follow-up (range, 5.2–34.3 months). Four patients experienced progression between 7.9 and 16.6 months after RT (PD group), and the nine presented no evidence of disease (NED group). The Dmax, the day with the highest ctDNA level among D0–7, was significantly different between the groups with total GE and max GE (*p* = 0.035 and 0.021, respectively). According to the ROC curves, the max GE showed the best AUC (86.1%) and the cut-off value of the Dmax was 1.5 (sensitivity: 66.7%, specificity: 100%, positive-predictive value: 100%, and negative-predictive value: 57.1%). Tumor size ≥ 3 cm, squamous histology, and a daily dose 3–4 Gy were correlated with the Dmax = D2–3. The Dmax showed better disease control rate with marginal significance (*p* = 0.081). **Conclusions:** The timing of early ctDNA elevation may have the potential to predict RT response. The max GE may be an index to verify the ctDNA levels after RT.

## 1. Introduction

Owing to advances in radiation therapy (RT) techniques—particularly in modalities such as intensity-modulated radiation therapy (IMRT) and image-guided radiation therapy (IGRT)—the ability to deliver highly conformal radiation doses while minimizing exposure to surrounding healthy tissues has significantly improved [1]. These techniques not only enhance local control but also reduce treatment-related toxicity, making them increasingly adopted in clinical practice across various cancer types. However, despite these technical improvements, the personalization of RT based on intrinsic tumor radiosensitivity remains a persistent and unmet challenge for radiation oncologists [2,3,4]. There is growing interest in identifying reliable biomarkers that could help tailor radiation treatment according to individual tumor biology.

Liquid biopsy has emerged as a promising tool in this regard, offering a non-invasive and repeatable method to analyze tumor-derived genetic material from blood samples. This approach enables clinicians to assess tumor-specific biomarkers, including circulating tumor DNA (ctDNA), in real time. Such applications can be integrated into RT protocols for various purposes, such as prognostication, risk stratification, monitoring of early tumor response, and detection of minimal residual disease (MRD) [5,6,7,8,9,10]. The minimally invasive nature of liquid biopsy allows longitudinal sampling, which is especially valuable in the context of RT, where treatment effects evolve over time.

Irradiation-induced tumor cell death is considered a major source of cell-free DNA (cfDNA) in circulation [11,12]. Consequently, an early elevation in cfDNA levels shortly after radiation exposure could serve as a potential surrogate marker for RT-induced tumor necrosis or cell lysis [12]. In our previous study, we demonstrated that irradiation significantly enhanced the sensitivity for detecting ctDNA in a xenograft mouse model, suggesting that radiation may promote the release of tumor DNA into the bloodstream [13]. Furthermore, in a preliminary clinical observation, a transient elevation in ctDNA levels within the first 72 h of RT initiation was detected in patients with lung cancer [13]. Although the initial findings were limited by a short follow-up period and lacked robust clinical outcome data, they offered a rationale for further exploration. In the present study, we aimed to more thoroughly investigate the early dynamic changes in ctDNA following the initiation of RT and to evaluate their potential association with clinical outcomes.

## 2. Materials and Methods

### 2.1. Patients

Two prospective studies were conducted once approval was obtained from the Institutional Review Board at the authors’ institution (approval no. 2017-09-120 and 2018-05-155). The first protocol (no. 2017-09-120) included patients with stage I–III non-small-cell lung cancer (NSCLC) according to the 8th American Joint Committee on Cancer Staging manual and pathological confirmation of their state as medically inoperable. The second protocol (no. 2018-05-155) included patients clinically diagnosed with lung cancer who failed to have a pathological diagnosis, for whom diagnostic procedures posed potential risks such as severe complications (e.g., hemorrhage) associated with invasive procedures, or cases where patients’ comorbidities (e.g., pulmonary fibrosis) increased the risk of adverse outcomes. Both studies were designed with a pilot format, aiming to enroll a maximum of 20 patients. All participants provided written informed consent prior to enrollment.

### 2.2. Radiotherapy

The patients underwent curative RT alone with 60–64 Gy in 4–20 fractions, and dose-fractionations were determined based on the tumor size, location, and stage [14]. For peripheral lung tumors, stereotactic body RT (SBRT) with 60 Gy in 4 fx was generally administered. For lesions that were not centrally located but were adjacent to the pleura, a regimen of 60–64 Gy in 8–15 fractions was considered to mitigate potential RT-related toxicities such as chest wall fibrosis or rib fracture. For centrally located lesions, most patients received 60 Gy in 20 fractions. Based on four-dimensional simulation computational tomography (CT), delineation of gross tumor volume (GTV) was performed. The clinical target volume (CTV) was expanded by 0–5 mm from the internal target volume (ITV) and subsequently adjusted with anatomical barriers. The planning target volume (PTV) was generated by applying a 5 mm margin from the CTV. Generally, RT plans were prescribed with 97% coverage for the PTV. As for the RT techniques, either IMRT or SBRT was used with 6- or 10-megavoltage Linac machines (TrueBeam or VitalBeam, Varian Medical System, Palo Alto, CA, USA). In some selected patients with poor pulmonary function or pre-existing lung conditions such as pulmonary fibrosis, proton therapy was considered, and the intensity-modulated proton therapy technique was used (Sumitomo Heavy Industries system, Chiba, Japan). As part of regular follow-up, the patients visited the hospital 1 month after RT, followed by visits every 3 months for chest CT. At every visit, treatment response evaluation was performed with RECIST criteria 1.1.

### 2.3. Blood Samples

The baseline peripheral blood samples (10 mL) were obtained on the day of the simulation for RT planning, with most being collected within a week prior to the start of RT (D0). In addition, serial sampling was performed on days 1, 2, and 3 (D1, D2, and D3, respectively) and 1 week (D7) from starting RT.

### 2.4. ctDNA

The processes related to ctDNA have been extensively detailed in our previous work, including nucleic acid extraction, quantification, library preparation, sequencing data processing, and detection of somatic mutations [13].

The ctDNA was extracted from plasma using the QIAamp Circulating Nucleic Kit (Qiagen, Santa Clara, CA, USA). Genomic DNA (gDNA) was isolated from blood samples using the QIAamp DNA Mini Kit (Qiagen). DNA and RNA concentrations were quantified using a Nanodrop 8000 UV-Vis spectrometer (Thermo Fisher Scientific, Waltham, MA, USA) and a Picogreen fluorescence assay on a Qubit 2.0 fluorometer (Life Technologies, Waltham, MA, USA). Fragment size distribution was analyzed using a 2200 TapeStation Instrument (Agilent Technologies, Santa Clara, CA, USA). The cfDNA was quantified by amplifying the LINE-1 locus using real-time PCR with SYBR Green. PCR primers for human LINE-1 were used to quantify cfDNA level. Data were analyzed using LightCycler 480 software (Roche, Branchburg, NJ, USA). The gDNA was a sonicated 150–200 bp fragment. Tumor biopsy libraries were prepared using the SureSelect XT reagent kit (Agilent Technologies) and the KAPA Hyper Prep Kit (Kapa Biopsystems, Woburn, MA, USA) for plasma DNA. Hybrid selection targeted ~117 kb of the genome, including 38 cancer-related genes. Whole exome sequencing libraries were generated using the SureSelect XT Human All Exon V5 kit (Agilent Technologies) and sequenced on an Illumina HiSeq 2500 system (Illumina, San Diego, CA, USA). Liquid biopsy data were aligned to the gh19 reference genome using BWA-mem (v0.7.5). Somatic mutations were identified with filtered steps for base quality, depth, and variant frequency, with further analysis using Stelka2.

### 2.5. Statistical Analysis

In our previous study, we used variant allele frequency (VAF, %) as a fundamental measure for quantifying ctDNA levels. In addition, genome equivalents (GE, hGE/mL) per plasma mL were calculated based on the product of the mean VAF and the plasma cfDNA concentration in each subject, divided by 0.0033 ng, which represents the estimated weight of a haploid human genome [15,16]. To identify a feasible index associated with progression, comprehensive variables quantifying various types of ctDNA, including sum of VAF from all detected variants (total VAF), maximum value of VAF in detected variants (max VAF), sum of GE (total GE), and maximum value of GE (max GE), were used as statistics. Also, the ratio of each measurement was defined as the value on a specific day divided by the value at D0.

The temporal dynamics of ctDNA were statistically evaluated using a generalized linear mixed (GLM) model to account for both fixed and random effects and to accurately reflect intra-individual variability over time. This model enabled the analysis of repeated ctDNA measures for each patient and helped identify patterns of ctDNA fluctuation during the first week of RT. Comparisons between patient subgroups were assessed using *t*-tests.

The time point with the most prominent increase in ctDNA levels during the observation window (days 0 through 7) was defined as Dmax. In addition, receiver operating characteristic (ROC) curve analysis was applied to assess the predictive accuracy of each ctDNA variable, and corresponding cut-off values for disease progression were determined based on the Youden index. Survival analysis was carried out using Kaplan–Meier methods to analyze the disease control rate.

All statistical analyses were performed using two independent software packages: R version 4.0.3 (R Foundation for Statistical Computing, Vienna, Austria, http://www.R-project.org (accessed on 1 January 2023)) for modeling and visualizations and SPSS version 27.0 (IBM Corp., Armonk, NY, USA) for descriptive statistics, group comparisons, and survival analysis. Statistical significance was set at a two-sided *p*-value of <0.05 for all tests.

## 3. Results

Of the 24 enrolled patients initially considered for the study, 13 patients were included in the final analysis after excluding those who had undetectable ctDNA levels or lacked at least one ctDNA sample collected within the first three days following the initiation of RT.

Upon diagnosis, the microscopic tumor types were as follows: six patients had squamous cell carcinoma, three had adenocarcinoma, and four had undetermined tumor types. Among these patients, only one individual tested positive for an EGFR mutation, while the remaining patients either tested negative or had unknown mutation status due to insufficient molecular data or limited tissue samples.

During a median follow-up duration of 22.2 months (range—5.2 to 34.3 months), disease progression was observed in four patients. The progression-free interval for these patients ranged from 7.9 to 16.6 months after the completion of RT. Of the four cases, two patients experienced local or locoregional recurrence and two had lung-to-lung metastases. The remaining nine patients in the study showed no evidence of disease (NED) during the entire observation period. This dichotomy in clinical outcomes (PD vs. NED) served as the basis for subsequent correlation analysis with early ctDNA dynamics. Patient characteristics and treatment details for both groups are summarized in Table 1.

The mutation profiles of the 13 ctDNA-detected cases varied from one to multiple somatic mutations per patient (Figure 1). Of the genes in which mutations were detected, mutations in *TP53*, *RYR2*, and *GATA3* were the most frequent, whereas mutations in *ERBB2*, *EGFR*, *KRAS*, *HRAS*, and *PTEN* were each observed once in individual patients.

Next, trend analysis of four variables related to cfDNA and ctDNA, involving both absolute values and relative values based on the baseline, was conducted. Since the sample did not exhibit a normal distribution, mixed analysis was attempted (Figure 2). The results revealed challenges in establishing meaningful interpretations for the absolute values within the first week following the initiation of RT. The NED group showed a higher ctDNA ratio (1.76–2.34) on D2 than the baseline level, although the difference was not significant (Figure 2G–J).

When comparing the NED and PD groups, *t*-tests for the maximum ratio on D2–3 and progression status were not significant (Figure 3A–E). However, Dmax was found to be significantly different between the NED and PD groups for the total GE and max GE (*p* = 0.035 and 0.021, respectively) (Figure 3G–J).

For the ROC curves, with Dmax, maxGE showed the best AUC (86.1%), and the cut-off value of Dmax was 1.5 (Figure 4). Based on this result, to predict NED, an increase in the maxGE value on D2–3 showed sensitivity, specificity, positive predictive, and negative predictive values of 66.7%, 100%, 100%, and 57.1%, respectively.

Next, we performed *t*-tests for subgroup analysis. Of the characteristics and Dmax, only Dmax was found to be related to PD (*p* = 0.015). However, a tumor size ≥ 3 cm, squamous histology, and a daily dose of 300–400 cGy were correlated to the Dmax of maxGE (*p* = 0.031, 0.031, and 0.031, respectively) (Figure 5).

The results of survival analysis indicated that none of the characteristics were significant. Despite this, Dmax showed different disease control rates with marginal significance (*p* = 0.081, Figure 6).

## 4. Discussion

ctDNA refers to small fragments of DNA that are released into the bloodstream from tumor cells [17]. It serves as a non-invasive biomarker for detecting genetic mutations and alterations associated with cancer [18]. While numerous studies have provided substantial evidence supporting the ability of ctDNA to predict treatment responses, the majority of studies have focused on the long-term observation of ctDNA dynamics [19,20,21]. However, RT-induced cell and DNA damage is known to occur rapidly after RT and is transmitted to the immune system within a few days [22]. The combination of immunotherapy and RT, administered almost concurrently, is known to be synergistic and has demonstrated clinical efficacy in multiple trials [23]. Therefore, early responses in ctDNA dynamics related to RT may be a crucial indicator for predicting response to treatment.

The precise timing of an increase in ctDNA following RT for NSCLC remains unclear. Some studies have reported no increase in ctDNA 3 and 5 days after RT [24], while others have observed a rapid increase in ctDNA as early as 2 h after RT initiation [25]. A significant increase in ctDNA has been reported after a single fraction or within 24–48 h after RT [26,27]. Similarly, in this study focusing on first-week ctDNA dynamics, we observed the most significant increase in ctDNA on the second day of RT, consistent with the results of a previous study [13].

While previous studies have observed an early increase in ctDNA, the specific impact of this phenomenon on clinical treatment outcomes remains incompletely understood. In the present study, six out of thirteen patients exhibited increased ctDNA ratios on the second or third day of treatment initiation, and none of these patients experienced recurrence within several years. The maxGE ratio ctDNA marker showed the best predictive value associated with NED. Although further research is needed to validate these observations, our findings suggest that maxGE, encompassing various ctDNA types, may be a useful indicator for predicting RT response and stratifying patient outcomes in clinical settings.

Subgroup analysis indicated that the day with the highest ctDNA ratio of maxGE was correlated with specific characteristics, including large tumor size, squamous histology, and a moderately hypofractionated dose. Generally, large tumors would be expected to result in a poorer treatment response [28]. However, our results did not align with these expectations, as these characteristics individually did not reach statistical significance in survival analysis. Although the interpretation of these results is challenging, owing to limited sample size and evidence, previous research on the combination of RT and immunotherapy indicating an association with tumor burden or hypofractionated dose seems consistent with our findings [29]. Further stratified analysis in larger cohorts is needed to elucidate these trends.

With the integration of RT with immunotherapy, pulsed RT, such as personalized ultrafractionated stereotactic adaptive radiotherapy (PULSAR), has recently gained increasing interest [30]. Repeated peaks in the levels of ctDNA with intermittent radiation exposure over several weeks may be more beneficial than relatively short-period conventional dose schedules, potentially providing a more favorable stimulation of the immune system [31]. Our findings may support the concept of PULSAR therapy by suggesting that fluctuations in ctDNA dynamics early during treatment could act as a real-time biomarker of systemic immune activation. Further research in this area, particularly integrating ctDNA monitoring into adaptive treatment protocols, is needed.

There were several limitations in this study. The sample size was small, hindering the presentation of conclusive results. Additionally, the absolute and relative values of ctDNA did not demonstrate significance. This may be attributed to substantial inter-individual variability in ctDNA levels and responses, influenced by factors such as tumor burden, cell type, and radiation dose. Furthermore, various DNA mutations were detected in each patient, preventing individual DNA-specific analyses. There are still uncertainties regarding the relationship between RT and ctDNA. We hope that the results of ongoing studies, such as the VIGILANCE trial (NCT06086574), will provide answers to these questions [32].

## 5. Conclusions

The timing of early ctDNA elevation after RT may serve as a potential predictive marker for a favorable clinical response. Based on our observations, we propose that maxGE could be a feasible and clinically applicable index for assessing ctDNA levels after RT. Further research with larger sample sizes, longitudinal designs, and comprehensive analysis of individual genomic alterations is warranted to validate these findings and better understand the complex interplay between ctDNA dynamics, tumor characteristics, treatment response, and immune modulation in the context of RT for NSCLC.

## Figures and Tables

**Figure 1 diagnostics-15-01198-f001:**
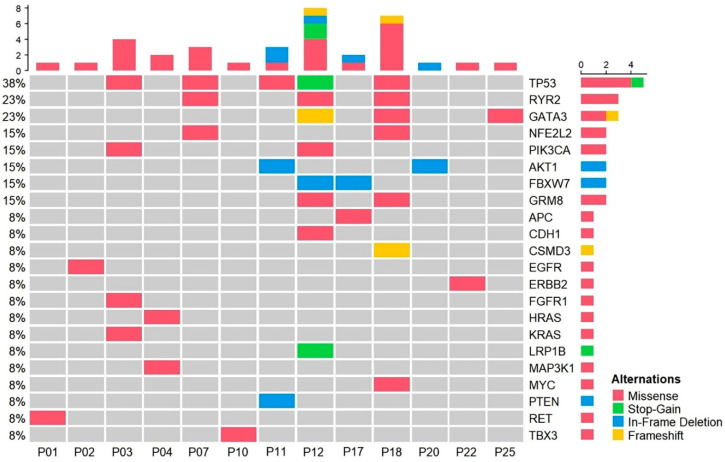
Mutation profiling of plasma samples from patients with radiotherapy. OncoPrint chart shows the occurrence of mutations as profiled by targeted ultra-deep sequencing techniques across 13 patients with radiotherapy.

**Figure 2 diagnostics-15-01198-f002:**
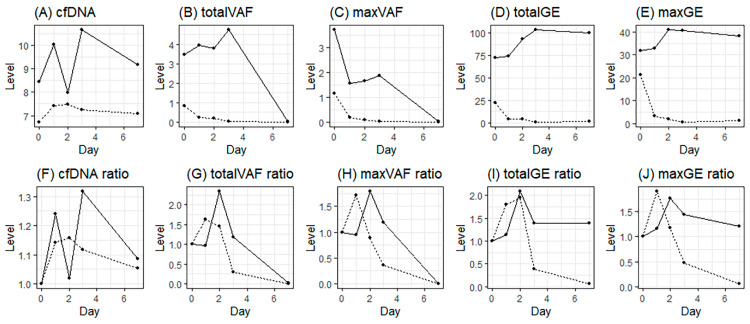
Daily ctDNA levels from the start of radiotherapy; (**A**–**E**), absolute mean values of cfDNA, total VAF, max VAF, total GE, and max GE; (**F**–**J**), relative mean values (ratio) from the baseline of cfDNA, total VAF, max VAF, total GE, and max GE (solid line, NED; dotted line, PD).

**Figure 3 diagnostics-15-01198-f003:**
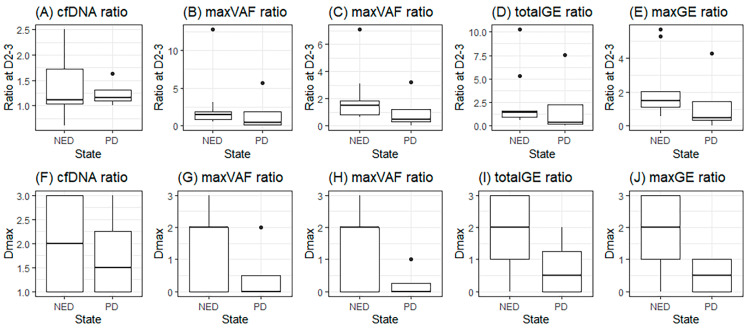
Dmax in groups with no evidence of disease (NED) or progression (PD); (**A**–**E**), maximum ratio during D2–3 of cfDNA, total VAF, max VAF, total GE, and max GE; (**F**–**J**), the day with maximum ratio (Dmax) of cfDNA, total VAF, max VAF, total GE, and max GE.

**Figure 4 diagnostics-15-01198-f004:**
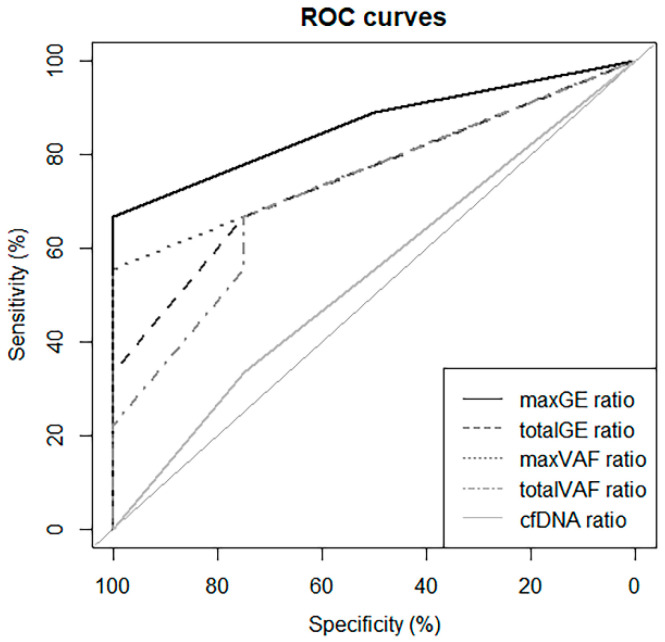
ROC curve using Dmax to predict progression.

**Figure 5 diagnostics-15-01198-f005:**
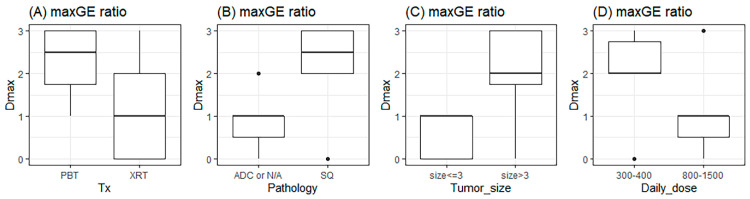
The relationship of Dmax with types of (**A**) treatment, (**B**) pathology, (**C**) tumor size, and (**D**) daily RT dose. The unit of tumor size and daily dose are cm and cGy. Tx: treatment; PBT: proton beam therapy; XRT: X-ray beam therapy; ADC: adenocarcinoma; N/A: not available; and SQ: squamous cell carcinoma.

**Figure 6 diagnostics-15-01198-f006:**
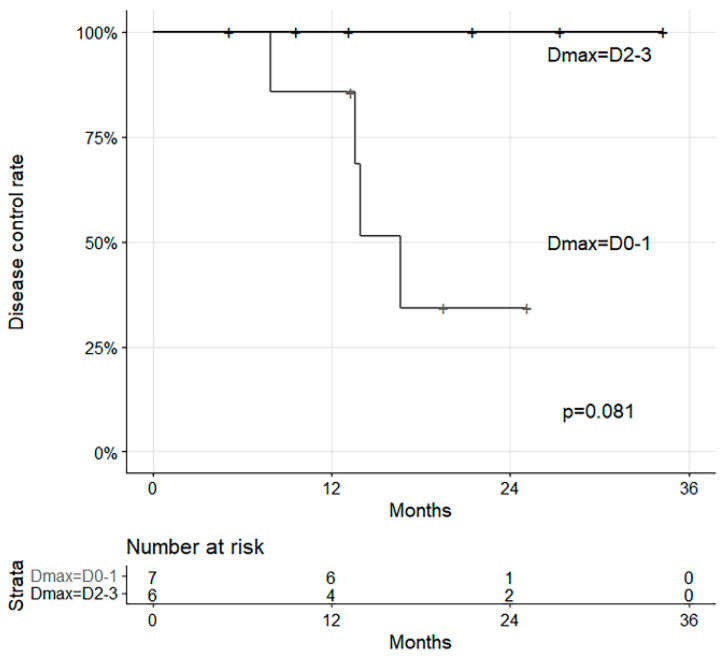
Survival curves with disease control and Dmax.

**Table 1 diagnostics-15-01198-t001:** Characteristics of patients and progression.

Patient No.	Age (Years)	Sex	Clinical Stage (AJCC 8th)	Tumor Size (cm)	Microscopic Tumor Type	Mutation	RT Type	RT Dose, Total (cGy)	RT Dose, Daily (cGy)	DFI (Months)	Progression Site(s)
01	78	Male	T2bN1M0, IIB	4.8	SQ	N/A	X-ray	6000	300	9.7	NED
02	75	Male	T2aN0M0, IB	3.3	SQ	N/A	Proton	6000	400	34.3	NED
03	82	Male	T1bN0M0, IA	1.9	N/A	N/A	X-ray	6400	800	13.9	Lung-to-lung
04	78	Female	T2aN0M0, IB	3.7	AD	EGFR(+) *	X-ray	6000	400	13.2	NED
07	62	Female	T4N0M0, IIIA	4.6	SQ	EGFR(−) ALK(−)	X-ray	6000	400	27.4	NED
10	63	Male	T1bN0M0, IA	1.9	N/A	N/A	Proton	6400	800	16.6	Lung-to-lung
11	74	Male	T2aN0M0, IB	5.2	SQ	EGFR(−) ALK(−)	Proton	6000	400	5.2	NED
12	75	Male	T3N0M0, IIB	3.1	SQ	EGFR(−) ALK(−)	Proton	6400	800	21.5	NED
17	71	Male	T1cN0M0, IA	3.1	SQ	N/A	X-ray	6400	800	7.9	Local PD
18	57	Male	T2aN0M0, IB	4.4	N/A	N/A	X-ray	6000	1500	25.2	NED
20	70	Male	T1bN0M0, IA	1.6	AD	EGFR(−) ALK(−)	X-ray	6400	800	19.6	NED
22	72	Male	T1bN0M0, IA	1.8	AD	N/A	X-ray	6400	800	13.3	NED
25	68	Male	T1cN0M0, IA	2.8	N/A	N/A	X-ray	6000	400	13.6	Locoregional PD

* The patient had pathologically reported mutations of EGFR, missense mutation of exon 21. Other patients had no mutation of EGFR and ALK or no information of specific mutations. RT: radiotherapy; DFI: disease-free interval; SQ: squamous cell carcinoma; AD: adenocarcinoma; EGFR: epidermal growth factor receptor; ALK: anaplastic lymphoma kinase; N/A: not available; PD: progressive disease; NED: no evidence of disease.

## Data Availability

The data presented in this study are available on request from the corresponding author due to privacy.

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
