# Peer review of "Early Dynamics of Circulating Tumor DNA Following Curative Hypofractionated Radiotherapy Related to Disease Control in Lung Cancer"

_diagnostics, 2025, doi:10.3390/diagnostics15101198_

Round 1
Reviewer 1 Report
Comments and Suggestions for Authors
Please note that reviewing reports are confidential.
The authors report data on circulating tumor DNA during radiotherapy for lung carcinomas. The topics is of major interest, revision would result in improvement.
The authors could add informations on the specimens that were used for pathology/microscopy diagnosis.
Do the authors consider that the ctDNA from the blood-sample taken at radiotherapy simulation was before the beginning of the radiotherapy treatment?
The authors could discuss in detail the biological significance of circulant tumor DNA: results from circulant tumor cells/emboli? other? The relationships between the ctDNA evolution and the presence of radiotherapy-related intratumor changes (ex necrosis) could also be presented/discussed. The authors could also mention the possible risk of post-radiotherapy metastases (possible association to ctDNA levels? tumor histotype? other).
Examples of words/phrases to revise
"definitive RT" can be reformulated
"can be incorporated into various RT strategies" can be reformulated, do the authors mean that liquide biopsy evaluation can be used for radiotherapy protocols/plans?
"cell killing" should be reformulated, do the authors mean "cell necrosis/apoptosis"?
"and determine the clinical relevance" can be reformulated "and evaluate the potential clinical relevance" similar
"were risky" should be reformulated by using scientific terms/phrases
"previous paper" can be reformulated "previous article/work" similar
""survival analysis" the authors could precise if overall survivial.
Table 1 "Pathology" can be changed to "Microscopy tumor type" similar
"sample did not exhibit" can be reformulated "the sample did not show"
"better disease control" can be reformulated
"poorly understood" can be reformulated "incompletely understood" similar
"out results contradicted these expectations" can be reformulated
"did not reach significance" the authors could prcise the type of significance: statistical? biological? other.
"increasing attention" can be changed to "increasing interest" similar
"are thought" can be changed
"There were some limitations in this study" should be reformulated
"unclear aspects" can be reformulated.
Comments on the Quality of English LanguageCan be improved.
Author Response
|
1. Summary |
|
|
|
Thank you very much for taking the time to review this manuscript. Please find the detailed responses below and the corresponding revisions in track changes in the re-submitted files.
|
||
|
2. Point-by-point response to Comments and Suggestions for Authors |
||
|
Comments 1: The authors could add information on the specimens that were used for pathology/microscopy diagnosis.
|
||
|
Response 1: Thank you for pointing this out. In Table 1, the mutation status of the specimens used for pathology and microscopy diagnosis is provided, detailing the identified mutations.
|
||
|
Comments 2: Do the authors consider that the ctDNA from the blood-sample taken at radiotherapy simulation was before the beginning of the radiotherapy treatment?
|
||
|
Response 2: Yes, the ctDNA was collected prior to the start of the radiotherapy treatment. This is because the simulation CT scan was performed within one week before the initiation of the RT treatment. The below sentence is added in the manuscript. è (Materials and methods) The baseline peripheral blood samples (10 ml) were obtained at the day of the simulation for RT planning, with most being collected within a week prior to the start of RT.
|
||
|
Comments 3: The authors could discuss in detail the biological significance of circulating tumor DNA: results from circulating tumor cells/emboli? other? The relationships between the ctDNA evolution and the presence of radiotherapy-related intratumor changes (ex necrosis) could also be presented/discussed.
|
||
|
Response 3: ctDNA is a distinct concept from circulating tumor cells/emboli, as it refers to the DNA fragments extracted from plasma in the blood. It reflects cell death or necrosis, which is relevant to tumor progression and response to therapy. We will add references to support the biological significance of ctDNA. è (Discussion) The ctDNA refers to small fregments of DNA that are released into the bloodstream from tumor cells. It serves as a non-invasive biomarkers for detecting genetic mutations and alterations associated with cancer. |
||
|
|
||
|
3. Additional clarifications |
||
|
Examples of words/phrases to revise:
|
||
"definitive RT" can be reformulated
"can be incorporated into various RT strategies" can be reformulated, do the authors mean that liquid biopsy evaluation can be used for radiotherapy protocols/plans?
"cell killing" should be reformulated, do the authors mean "cell necrosis/apoptosis"?
"and determine the clinical relevance" can be reformulated "and evaluate the potential clinical relevance" similar
"were risky" should be reformulated by using scientific terms/phrases
"previous paper" can be reformulated "previous article/work" similar
"survival analysis" the authors could precise if overall survival.
Table 1 "Pathology" can be changed to "Microscopy tumor type" similar
"sample did not exhibit" can be reformulated "the sample did not show"
"better disease control" can be reformulated
"poorly understood" can be reformulated "incompletely understood" similar
"our results contradicted these expectations" can be reformulated
"did not reach significance" the authors could precise the type of significance: statistical? biological? other.
"increasing attention" can be changed to "increasing interest" similar
"are thought" can be changed
"There were some limitations in this study" should be reformulated
"unclear aspects" can be reformulated.
Response 4: We greatly appreciate the detailed comments. We have positively incorporated all of them and made the revisions.
Reviewer 2 Report
Comments and Suggestions for Authors
The manuscript entitled” Early Dynamics of Circulating Tumor DNA Following Curative Hypofractionated Radiotherapy Related to Disease Control in Lung Cancer” evaluated early changes in ctDNA levels during the first week of hypofractionated RT in lung cancer, and assess their potential as biomarkers for predicting treatment response. It explores a novel time window (within the first 7 days of RT) which has not been extensively studied in previous research.
1.Some grammatical errors and awkward phrasings (e.g., Abstract line 29: “may possibility to predict” should be “may have the potential to predict”).
2.Consider streamlining sections with technical details (e.g., mutation profile) by referencing previous work more concisely.
3. Emphasize more in the Introduction why early detection of treatment response is essential—especially compared to conventional imaging or long-term follow-ups.
Author Response
|
1. Summary |
|
|
|
Thank you very much for taking the time to review this manuscript. Please find the detailed responses below and the corresponding revisions/corrections highlighted/in track changes in the re-submitted files.
2. Point-by-point response to Comments and Suggestions for Authors |
||
|
Comments 1: Some grammatical errors and awkward phrasings (e.g., Abstract line 29: “may possibility to predict” should be “may have the potential to predict”).
|
||
|
Response 1: Thank you for pointing this out. We agree with this comment and change the sentence at the abstract.
|
||
|
Comments 2: Consider streamlining sections with technical details (e.g., mutation profile) by referencing previous work more concisely.
|
||
|
Response 2: We have included the essential technical contents from the previous study.
|
||
|
Comments 3: Emphasize more in the Introduction why early detection of treatment response is essential—especially compared to conventional imaging or long-term follow-ups.
|
||
|
Response 3: In the introduction, we presented the results and insights gained from our previous study, which inspired the current research. The corresponding details are discussed in the discussion section.
|
||